# Making Steppingstones out of Stumbling Blocks: A Bayesian Model Evidence Estimator with Application to Groundwater Transport Model Selection

**Ahmed S. Elshall [1,2]** and **Ming Ye [3,*]**

[1]  Department of Earth Sciences, University of Hawai'i Manoa, Honolulu, HI 96822, USA
[2]  Water Resources Research Center, University of Hawai'i Manoa, Honolulu, HI 96822, USA
[3]  Department of Earth, Ocean, and Atmospheric Science, Florida State University, Tallahassee, FL 32306, USA
[*]  Correspondence: mye@fsu.edu; Tel.: +850-644-4587; Fax: +850-644-0098

**Abstract:** Bayesian model evidence (BME) is a measure of the average fit of a model to observation data given all the parameter values that the model can assume. By accounting for the trade-off between goodness-of-fit and model complexity, BME is used for model selection and model averaging purposes. For strict Bayesian computation, the theoretically unbiased Monte Carlo based numerical estimators are preferred over semi-analytical solutions. This study examines five BME numerical estimators and asks how accurate estimation of the BME is important for penalizing model complexity. The limiting cases for numerical BME estimators are the prior sampling arithmetic mean estimator (AM) and the posterior sampling harmonic mean (HM) estimator, which are straightforward to implement, yet they result in underestimation and overestimation, respectively. We also consider the path sampling methods of thermodynamic integration (TI) and steppingstone sampling (SS) that sample multiple intermediate distributions that link the prior and the posterior. Although TI and SS are theoretically unbiased estimators, they could have a bias in practice arising from numerical implementation. For example, sampling errors of some intermediate distributions can introduce bias. We propose a variant of SS, namely the multiple one-steppingstone sampling (MOSS) that is less sensitive to sampling errors. We evaluate these five estimators using a groundwater transport model selection problem. SS and MOSS give the least biased BME estimation at an efficient computational cost. If the estimated BME has a bias that covariates with the true BME, this would not be a problem because we are interested in BME ratios and not their absolute values. On the contrary, the results show that BME estimation bias can be a function of model complexity. Thus, biased BME estimation results in inaccurate penalization of more complex models, which changes the model ranking. This was less observed with SS and MOSS as with the three other methods.

**Keywords:** groundwater transport; Bayesian; model selection; Bayesian model averaging; modeling; model complexity; Bayesian model evidence; marginal likelihood

## 1. Introduction

Bayesian statistics is gaining popularity in hydrological modeling (e.g., [1–6]). It is an appealing choice for ranking candidate conceptual models [7–10], modeling propositions [2,11–15] and scenarios [16,17]. Due to its sound logical foundation [18], Bayesian scientific reasoning is an appealing paradigm that brings the model up against data to let the data speak based on the principle of parsimony. The theory can naturally entertain multiple working hypotheses [19] such that scientific and modeling propositions with good empirical evidence will stand out. This *evidence*, which is also

interchangeably written as *model evidence* [20], or *Bayesian model evidence* (BME) [21] is of fundamental importance in Bayesian multi-model analysis. It states the overall probability of the model to reproduce the observation data given all the parameter values that the model can assume, and thus permits the comparisons of candidate models.

Bayesian inference provides a quantitative measure of the degree of belief of a model $M_k$ being true given data **D**. This is started by exploring the posterior probability distribution $p(\theta_k|\mathbf{D}, M_k)$ of the parameters of interest $\theta_k$

$$p(\theta_k|\mathbf{D}, M_k) = \frac{p(\mathbf{D}|\theta_k, M_k)p(\theta_k|M_k)}{p(\mathbf{D}|M_k)} \tag{1}$$

with the Bayesian model evidence (BME) being

$$p(\mathbf{D}|M_k) = \int p(\mathbf{D}|\theta_k, M_k)p(\theta_k|M_k)d\theta_k \tag{2}$$

such that $p(\mathbf{D}|\theta_k, M_k)$ is the likelihood of model $M_k$ and its parameter set $\theta_k$ in reproducing the observation data **D**, and $p(\theta_k|M_k)$ is the prior densities of parameters $\theta_k$ under model $M_k$. The likelihood $p(\mathbf{D}|\theta_k, M_k)$ states how likely the model would reproduce the observation data for a given realization of the model parameters. By marginalizing out the model parameters, the BME $p(\mathbf{D}|M_k)$ states how likely the model would reproduce the observation data given all the parameter values that the model can assume. Thus given the same observation data **D**, the BME is used for evaluating and ranking candidate models by their ability to reproduce the observation data. Note that the BME is also referred to in literature as *marginal likelihood* [22–24] , *integrated likelihood* [25–27], *normalizing constant* [28], *marginal evidence* [15] , among other names. This research is of important relevance to hydrology since hydrologic systems are complex natural systems, and thus we often deal with multiple conceptual models. Multi-model analysis techniques such as Bayesian model averaging and Bayesian model selection are becoming very popular in hydrology. Estimating BME is an essential step in model diagnostics and assessment with respect to model complexity [29], and thus is shared among these multi-model techniques. The reader is referred to a recent review article [30] that discusses BME in relation to multi-model analysis in hydrology.

The BME can be exactly evaluated using an analytical solution, approximated using a semi-analytical solution, or numerically estimated using Monte Carlo simulation. Although exact and fast, analytical solutions are only available for simple models with prior, likelihood, and posterior being Gaussian distributions [22–24]. Yet for practical applications in hydrology, semi-analytical solutions are most commonly used [7,9,10,12–14,21,26,31–41]. Semi-analytical solutions include Laplace approximations [42] and approximate information criteria such as Bayesian information criterion (BIC) [43] and Kashyap information criterion (KIC) [44].

Semi-analytical solutions have two main limitations. First, these methods have different simplifying assumptions and are subject to truncation error [24]. This can lead to inaccurate estimation of the BME, resulting in contradicting results with respect to model ranking [21,26,36,38]. A second limitation of these methods is that they do not explicitly account for the impact of the prior distribution. Evaluating the BME is an important tool for model complexity analysis [29,45–49]. The main purpose for using BME as a model ranking criterion is not only to rank the candidate models based on their goodness-of-fit as this can be done by much simpler metrics, such as root mean squared error or Nash–Sutcliffe model efficiency, but also to penalize the model with higher complexity. Semi-analytical solutions such as BIC indirectly penalize model complexity through assigning penalties for each parameter regardless of its prior distribution. Other methods such as KIC and Laplace approximations would consider the Fisher information matrix to account for a prior. Accordingly, the prior distribution is not considered in strict Bayesian definition, which is that BME is the weighted average of all the likelihood values that model can assume with weights coming from the prior (Equation (2)). In other words, as the low-likelihood region in the parameter space gets wider, the BME should become lower. For example, if we used an infinite uniform prior distribution the BME will approach zero. The Fisher

information matrix does not strictly account for the prior distribution according to this definition. Thus, semi-analytical solutions do not consider the prior distribution in a strict Bayesian sense. Yet it is favorable sometimes to penalize model complexity by specifying the penalty of each individual parameter through considering its prior distribution. In other words, the BME is expected to be low for peaked likelihood, wide non-informative prior and high-dimensional prior. Alternatively, the BME is high when the likelihood and the prior distributions are concentrated over the same parameter region. Accordingly, the main advantage of the Monte Carlo simulation methods is the explicit accounting for the prior. Theoretical and numerical comparison of semi-analytical technique and Monte Carlo simulation technique for evaluating the BME is beyond the scope of this work, and the reader is referred to other recent studies [21,24,26].

To improve on the above limitations of semi-analytical solutions, numerical estimation of the BME with Monte Carlo based methods has become a fundamental computational problem in Bayesian statistics. Investigating the use of Monte Carlo methods to estimate BME has recently gained research interest in hydrology [21,24,26,45,48,50–52]. These studies leverage on the significant effort in different branches of science to develop robust BME numerical estimators with the aim of reducing the estimation bias and increasing the computational efficiency. We briefly review few of these estimators here. As indicated by Equation (2), numerical estimation of BME using Monte Carlo simulation is not a trivial task as it requires evaluating a multi-dimensional integral with a dimension equal to the number of model parameters. This can be done directly through integrating the likelihoods over the parameter space or indirectly through calculating the sample acceptance ratio. For example, Marshall et al. [35] used the Chib and Jeliazkov method [53] to estimate BME for comparing hydrological models. This method uses the acceptance of the Metropolis–Hasting sampler to estimate the BME. Friel and Wyse [20] reviewed BME estimation methods and pointed out that the Chib and Jeliazkov estimator [53] is generally accurate and exhibits very small dispersion for independent repeated runs. Although the method is flexible, Marshall et al. [35] study noted that it can be difficult to implement with efficient Markov chain Monte Carlo (MCMC) methods for complex or high-dimensional models. This is mainly because the method of Chib and Jeliazkov [53] requires the use of an MCMC algorithm with a single-chain proposal distribution to obtain the Metropolis–Hasting ratio. Other indirect BME estimation techniques that are based on a single-chain proposal distribution are annealed importance sampling [54], reversible-jump MCMC [55], and sequential Monte Carlo sampling [56,57]. Generally, single-chain MCMC algorithms are not as robust and convenient as multi-chain MCMC algorithms [58–61]. Multi-chain MCMC algorithms can avoid premature convergence and can better handle sampling difficulties such as high nonlinearity, multimodality, non-separablity, heavy tailed densities, and high-dimensionality [59–62]. In addition, designing the proposal distribution of single-chain MCMC algorithms is a model-specific and a time-consuming task that requires a lot of experience. Thus, it is desirable to pursue the direct likelihood-based BME estimation methods, which can be implemented with both single- and multi-chain MCMC schemes.

Direct likelihood-based methods for BME estimation employ ideas from importance sampling, nested sampling, and path sampling approaches. As the BME is the weighted average of the likelihoods where the weight comes from the prior, the most obvious solution is to sample the full prior distribution using brute-force Monte Carlo (MC) simulation. By the law of large numbers, this multidimensional integral (Equation (2)) can be estimated such that the BME is the weighted average of the likelihoods and weights come from the prior probability density function. This is known as the arithmetic mean estimator (AM) [63], which is an importance sampling approach ([64], p.134), such that the prior is the importance distribution. Although AM is straightforward to implement [27,65], it can be impractically computationally expensive to compute [21,24], and results in BME underestimation. This is especially true for problems with peaked likelihood and high dimensionality [22,24] as the high-likelihood regions are not adequately sampled. Alternatively, by using the posterior as our importance distribution, Newton and Raftery [66] show that the harmonic mean (HM) of the posterior sample likelihoods will converge to the BME. An HM estimator is generally used [14,67] as it comes at no extra computational

cost since the posterior distribution needs to be sampled anyway. However, an HM estimator is insensitive to the changes in the prior [20]. Accordingly, HM is susceptible to BME overestimation, especially for peaked likelihood [21,22,24]. To overcome the limitations posed by the prior sampling of AM estimator and the posterior sampling of HM estimator, Newton and Raftery [66] propose the stabilized harmonic mean method that uses an empirical parameter to develop a mixture of the prior and the posterior samples. Yet calibration of the empirical parameter is not straightforward [24] and the method still overestimates the BME [22] . HM can also be prone to numerical instability that can result in an enormous underestimation of the BME [20]. Moreover, few recent studies have shown that HM is a theoretically biased estimator [23,68]. However, recent advances in importance sampling methods have resulted in robust BME estimators [48].

To avoid the above limitations is to convert the multidimensional integral in Equation (2) to a one-dimensional integral, which can be easily integrated using any quadrature rule. This can be carried out through the semi-empirical method of nested sampling [69] and the path sampling method of thermodynamic integration [22,28,70]. The nested sampling method that is popular in astronomy was introduced to the hydrology community by Elsheikh et al. [50], and the thermodynamic integration method that is popular in phylogeny was introduced to the hydrology community by Schoups and Vrugt [45] and Liu et al. [24]. Nested sampling avoids sampling the full prior by converting the multidimensional integration of Equation (2) into a one-dimensional integral by relating the likelihood to the prior mass (i.e., integration of prior within a region). This is done through a nested procedure that requires local sampling instead of directly sampling from the prior to the posterior parameter space. Nested sampling is a semi-empirical procedure as the prior mass is empirically determined and involves heuristic terms. Although the method produces unbiased estimates [20,21,24] and is computationally efficient [24] , it exhibits high dispersion for repeated runs [20,24] and the estimation bias and dispersion grow with increasing the parameter dimension [24,71]. This deteriorated performance can be attributed to at least two reasons. First, the common implementation of nested sampling in hydrology uses the Metropolis–Hasting algorithm for local sampling [21,24,50], which is not robust for generating samples from complex sampling space. Nevertheless, the method is computationally general, and more robust MCMC sampling methods can be accommodated [51,52]. Second, as the construction of the prior mass is highly uncertain [69], the study of Lie et al. [24] highlighted that the nested sampling procedure does not guarantee that samples are systematically generated from the prior space to the posterior space. Yet unlike nested sampling, the path sampling method of thermodynamic integration (TI) samples the full prior.

Path sampling is a theoretically unbiased and a mathematically rigorous approach that systematically and directly samples intermediate parameter density functions spanning from the prior to the posterior parameter distributions. Given the distance between any two density functions and their corresponding expected densities, the path sampling method of TI converts the multidimensional integral in Equation (2) to a simple one-dimensional integral that can be easily integrated using any quadrature rule. Lie et al. [24] evaluated TI against semi-analytical solutions, AM, HM, and nested sampling and showed that TI outperformed the other methods in terms of solution accuracy and consistency for repeated independent runs. In addition, the groundwater model weights obtained using TI improved the predictive performance of Bayesian model averaging. However, the main drawback is that TI accuracy comes at a high computational cost [20,24]. To improve the computational efficiency of BME, while using a theoretically unbiased method, this study introduces the numerical estimation method of steppingstone sampling [23], which to our knowledge has not been attempted in hydrologic modeling. Steppingstone sampling (SS) utilizes a sampling path similar to TI to bridge from the prior to posterior distributions. The main idea of SS is that given any two adjacent parameter density functions, the one which is slightly more dispersed can act as an excellent importance distribution. This makes SS more computationally efficient as it requires fewer intermediate distributions than TI to bridging from the prior to the posterior while maintaining the same accuracy of BME estimation.

With application to groundwater transport modeling, in this study, we show that accurate estimation of the BME is important for accurate penalization of more complex models and accordingly, model selection. The manuscript has four objectives. First, by comparing SS with TI, the study shows SS improves on TI in terms of computational efficiency and tuning effort. SS requires fewer intervals to discretize the path from the prior to the posterior. In addition, TI is sensitive to the location of the discretization intervals, while SS is relatively invariant. However, these results are based on a Gaussian model where the prior, posterior, and intermediate distributions can be sampled directly without using MCMC, and thus without introducing any MCMC sampling error. Second, we further show that theoretically unbiased estimators such as TI and SS could have a large bias in practice arising from numerical implementation as MCMC sampling error can introduce bias. MCMC sampling error refers to an inaccurate approximation of the target stationary distribution. Third, we additionally introduce a variant of SS—namely, multiple one-steppingstone sampling (MOSS)—that is less susceptible to sampling error, and thus can potentially overcome shortcomings of TI and SS with respect to sampling error. Fourth, we evaluate these three numerical estimators of TI, SS, and MOSS along with the two limiting cases of AM and HM using a model selection problem of four groundwater transport models with different degrees of model complexity and model fidelity. Model complexity refers to the number of model parameters, and model fidelity refers to the degree of model realism with respect to the model structure, input data, and boundary conditions. By this example, we show that accurate BME estimation results in accurate penalization of more complex models. In addition, inaccurate BME estimation affects model weights and may accordingly change the model ranking. In summary, the objective of this research is to introduce the SS estimator, and show how BME estimation bias affects the penalization of model complexity differently for different estimators. Thus, the scope of this work is the accurate estimation of BME given different Monte Carlo estimators. Studying the impact of prior distribution [72], likelihood function [73,74] , model fidelity [12,14,15,47] , prior model probability [13], input data [75], and observation data [2,12] on the magnitude of BME is beyond the scope of this work. The readers are refered to a recent review article that discusses multi-model analysis in hydrology using Bayesian techiques [76].

The manuscript is organized as follows. In the Section 2, we introduce the five estimators and establish their theoretical connection with reference to importance sampling and path sampling techniques. Section 3 evaluates the accuracy, numerical efficiency, and tuning of the five estimators using a high dimensional problem, which has an analytical solution and is not subject to MCMC sampling error. In Section 4, we use multiple groundwater transport models to evaluate the impact of BME estimation bias on model complexity evaluation and model selection in the presence of MCMC sampling error. We summarize our main findings and provide recommendations in Section 5.

## 2. Methodology

### 2.1. Bayesian Model Averaging

Given a set of unobservable models $\mathbf{M} = (M_1, \ldots, M_K)$, the probability of the model given data $\mathbf{D}$ is referred to as the posterior model probability, or model weight. According to Bayes' rule [77] the weight $p(M_k|\mathbf{D})$ of any given model $M_k$ is

$$p(M_k|\mathbf{D}) = \frac{p(\mathbf{D}|M_k)p(M_k)}{\sum\limits_{l=1}^{K} p(\mathbf{D}|M_l)p(M_l)} \qquad (3)$$

such that the resulting model weight is based on the prior model probability $p(M_k)$ and the BME $p(\mathbf{D}|M_k)$, and is normalized by $\sum\limits_{l=1}^{K} p(\mathbf{D}|M_l)p(M_l)$ to make all model weights of the $K$ models add to unity. Prior model probability is our degree of belief about the model before it is conditioned on the data. Generally, the prior model probabilities are assigned equal values for all the models $p(M_k) = 1/K$,

but can be assigned differently for each model if there exists prior knowledge [13], or preference for any candidate conceptual model [78,79]. More importantly, BME (Equation (2)), measures the average fit of the model to data without reference to the model parameters. Note that as the BME estimation methods discussed below are applied to individual models, hereinafter we drop the subscript $k$ of model $M_k$ and parameter set $\theta_k$ for simplicity.

## 2.2. Path Sampling

Path sampling approach [28,80,81] is applied for a wide class of problems. Here we only focus on TI [22,70,82] for estimating the BME, which is a path sampling method. We consider two unnormalized power posterior densities $q_0(\theta)$ and $q_1(\theta)$, which are not necessarily from the same family (e.g., a normal distribution is in the exponential family), but with the same support (e.g., infinite support, or semi-infinite support). The goal of path sampling is to construct a continuous and differentiable path that links the two unnormalized power posterior densities using a scalar parameter $\beta \in [0,1]$ with

$$q_\beta(\theta) = p(\mathbf{D}|\theta, M)^\beta p(\theta|M) \tag{4}$$

that has a normalizing constant $Z_\beta$ to yield the normalized power posterior density

$$p_\beta(\theta) = \frac{q_\beta(\theta)}{Z_\beta} \tag{5}$$

where

$$Z_\beta = \int q_\beta(\theta)d\theta. \tag{6}$$

The power coefficients $\beta \in [0,1]$ discretize the power posterior parameter space starting from the prior to the posterior distributions. Using a thermodynamic analogy the scalar parameter $\beta \in [0,1]$ can be thought of as inverse temperature that tempers the likelihood surface such that power posterior $p_\beta(\theta)$ for $\beta = 1$ is equivalent to posterior distribution $p_1(\theta) = p(\mathbf{D}|\theta, M)p(\theta|M)/Z_1$; for $0 < \beta < 1$ we get any intermediate power posterior parameter space $p_\beta(\theta)$ as specified by $\beta$; and $\beta = 0$ is equivalent to the prior distribution $p_0(\theta) = p(\theta|M)/Z_0 = p(\theta|M)$. Note that $Z_0$ is the prior marginalized over $\theta$, which is equal to 1.

The goal is to estimate this ratio $Z_1/Z_0$, which can be represented by this integral

$$p(\mathbf{D}|M_k) = \frac{Z_1}{Z_0} = \exp(\log Z_1 - \log Z_0) = \exp\left(\int_0^1 \frac{\partial \log Z_\beta}{\partial \beta}d\beta\right) = \exp\left(\int_0^1 E_\theta[U(\theta)]d\beta\right) \tag{7}$$

and we define the *potential*

$$U(\theta) = \frac{\partial \log q_\beta(\theta)}{\partial \beta}. \tag{8}$$

The integral in Equation (7) is derived by taking the logarithm and differentiating the unnormalized power posterior $q_\beta(\theta)$ with respect to $\beta$, which according to Lartillot and Philippe [22] leads to

$$\begin{aligned} \frac{\partial \log Z_\beta}{\partial \beta} &= \frac{1}{Z_\beta}\frac{\partial Z_\beta}{\partial \beta} = \frac{1}{Z_\beta}\int \frac{\partial q_\beta(\theta)}{\partial \beta}d\theta \\ &= \int \frac{1}{q_\beta(\theta)}\frac{\partial q_\beta(\theta)}{\partial \beta}\frac{q_\beta(\theta)}{Z_\beta}d\theta = \int \frac{\partial \log q_\beta(\theta)}{\partial \beta}p_\beta(\theta)d\theta \\ &= E_\theta[\frac{\partial \log q_\beta(\theta)}{\partial \beta}] = E_\theta[U(\theta)] \end{aligned} \tag{9}$$

such that $E_\theta$ is the expectation with respect to $p_\beta(\theta)$, and then we integrate with respect to $\beta \in [0,1]$. It is convenient to obtain a closed-form expression for the potential $U(\theta)$. Thus, given Equation (8) and Equation (4) we obtain:

$$
\begin{aligned}
U(\theta) = \frac{\partial \log q_\beta(\theta)}{\partial \beta} &= \frac{1}{q_\beta(\theta)} \frac{\partial q_\beta(\theta)}{\partial \beta} \\
&= \frac{1}{q_\beta(\theta)} p(\mathbf{D}|\theta, M)^\beta \log p(\mathbf{D}|\theta, M) p(\theta|M) \\
&= \frac{1}{q_\beta(\theta)} q_\beta(\theta) \log p(\mathbf{D}|\theta, M) = \log p(\mathbf{D}|\theta, M)
\end{aligned}
\tag{10}
$$

Then we substitute Equation (10) into Equation (7) yielding the general form of the TI estimator of BME

$$
p(\mathbf{D}|M) = \exp \int_0^1 E_\theta[\log p(\mathbf{D}|\theta, M)]d\beta.
\tag{11}
$$

Thus, TI converts the multi-dimensional integral in Equation (2) to the simple one-dimensional integral in Equation (11), which can be estimated by many quadrature rules.

Solving Equation (11) requires determining the discretization (i.e., number and locations) of the power posterior coefficients $\beta \in [0,1]$ and selecting the quadrature rule for estimating $p(\mathbf{D}|M)$. For example, by defining $K+1$ equally spaced power coefficients $\beta \in [0,1]$, Lartillot and Philippe [22] used Simpson's triangulation. For the more general case of unequally spaced $\beta$, Friel and Pettit [70] use the trapezoidal rule for solving Equation (11). Accordingly, the TI estimator is:

$$
\hat{p}_{TI}(\mathbf{D}|M) = \exp\left(\int_{\beta_0}^{\beta_K} y(\beta)d\beta\right) \approx \exp\left(\sum_{k=1}^{K} (\beta_k - \beta_{k-1}) \frac{y_k + y_{k-1}}{2}\right)
\tag{12}
$$

such that $\beta_0 = 0$ and $\beta_K = 1$; and $y_k$ corresponding to $\beta_k$ is the average of log-likelihood:

$$
y_k = E_\theta[U_{\beta_k}(\theta)] = E_\theta[\log p_{\beta_k}(\mathbf{D}|\theta, M)] \approx \frac{1}{n} \sum_{i=1}^{n} \log p(\mathbf{D}|\theta_{\beta_k,i}, M)
\tag{13}
$$

where $\theta_{k,i}$ is a parameter sample from the power posterior distribution $p_{\beta_k}(\theta)$, and $p(\mathbf{D}|\theta_{k,i}, M)$ is the joint likelihood of that parameter sample $\theta_{k,i}$. The TI estimate is finally obtained by solving Equation (12) through drawing parameter samples and calculating their log-likelihoods $\ln p_{\beta_k}(\mathbf{D}|\theta, M)$ for each $\beta_k$ value. In case there is an analytical form for the power posterior distribution $p_{\beta_k}(\theta)$, we draw samples directly from them for different $\beta_k$ values using Monte Carlo simulation. This is not the case for most practical applications in hydrology, and MCMC is needed to estimate and draw samples from the unnormalized power posterior $q_{\beta_k}(\theta)$ distributions. Note that drawing parameter samples from the unnormalized power posterior $q_{\beta_k}(\theta)$ and the power posterior $p_{\beta_k}(\theta)$ distributions are equivalent.

The accuracy of TI depends on the number and values of $\beta_k$. The TI estimator becomes more accurate when the number of $\beta_k$ increases, such that in the absence of any numerical and sampling errors the TI estimator almost converges to the true solution when the number of $\beta$ is large enough. In practice, it is often desirable to optimize the $\beta$ values to reduce their number. It is desirable to place more $\beta$ values where the power posterior is changing rapidly (Friel and Pettitt, 2008; Calderhead and Girolami, 2009). Following [23] we use the positively skewed $Beta(\alpha, 1)$ distribution to assign the $\beta$ values such that

$$
\beta_k = (k/K)^{1/\alpha}
\tag{14}
$$

where $k = 1, 2, \ldots, K$. For example, as noted by Xi et al. (2011), setting $\alpha = 0.25$ provides the same $\beta$ schedule as [70] in which most $\beta$ values are placed near $\beta = 0$, while for $\alpha = 1$ the $\beta$ values are equally spaced, similar to [22]. For $\alpha = 0.3$ half of $\beta$ values will be less than 0.1, which is generally a reasonable choice as suggested by [23]. The number and values of $\beta_k$ can be also assigned manually through

an iterative procedure [24] , which is time-consuming as noted by [83]. It could be automatically tuned through using a derivative-based optimization method [82]. Yet this is beyond the scope of this manuscript.

*2.3. Importance Sampling*

The idea of importance sampling is to evaluate a distribution that we cannot sample from, while using samples from another distribution (i.e., the importance distribution in accordance with the importance weight). By sampling the importance distribution $g(\theta)$ the BME can be estimated using importance sampling [23]:

$$p(\mathbf{D}|M) = \frac{E_g[p(\mathbf{D}|\theta, M)w(\theta|M)]}{E_g[w(\theta|M)]} \approx \frac{\frac{1}{n}\sum_{i=n}^{n} p(\mathbf{D}|\theta_i, M)w_i(\theta_i|M)}{\frac{1}{n}\sum_{i=n}^{n} w_i(\theta_i|M)} \tag{15}$$

with

$$w(\theta_i|M) = \frac{p(\theta_i|M)}{g(\theta_i)} \tag{16}$$

such that $p(\mathbf{D}|\theta_i, M)$, $p(\theta_i|M)$, $w(\theta_i|M)$, and $g(\theta_i)$ are the joint likelihood, prior density, importance weight, and the importance density, respectively, computed at parameter sample $\theta_i$. For example, by choosing the prior $g(\theta) = p(\theta|M)$ as the importance distribution Equation (15) yields to the arithmetic mean (AM) [63] estimator:

$$\hat{p}_{AM}(\mathbf{D}|M) \approx \frac{1}{n}\sum_{i=1}^{n} p(\mathbf{D}|\theta_i, M) \tag{17}$$

where $p(\mathbf{D}|\theta_i, M)$ is the joint likelihood of sample $\theta_i$ drawn from the prior distribution $f(\theta|M)$. Alternatively, using the posterior $g(\theta) = f(\theta|\mathbf{D}, M)$ as the importance distribution yields the harmonic mean (HM) [66] estimator:

$$\hat{p}_{HM}(\mathbf{D}|M) \approx n / \sum_{i=1}^{n} \frac{1}{p(\mathbf{D}|\theta_i, M)} \tag{18}$$

where $p(\mathbf{D}|\theta_i, M)$ is the likelihood of sample $\theta_i$ drawn from the posterior distribution $p(\theta|\mathbf{D}, M)$.

Similarly, by using the power posterior distribution $g(\theta) = p_{\beta_{k-1}}(\theta)$ as previously defined in Equation (5) as the importance distribution, Equation (15) yields the steppingstone sampling (SS) [23] estimator:

$$p_{SS}(\mathbf{D}|M) \approx \prod_{k=1}^{K} \frac{1}{n}\sum_{i=1}^{n} p\left(\mathbf{D}|\theta_{k-1,i}, M\right)^{\beta_k - \beta_{k-1}} \tag{19}$$

which uses a sampling path similar to TI to bridge from the prior to posterior distributions with $K$ discrete $\beta$ values. Equation (19) reduces to AM estimator for the special case of $K = 1$. Note that $p(\mathbf{D}|\theta_{k-1,i}, M)$ is the joint likelihood of sample $\theta_{k-1,i}$ drawn from importance distribution $g(\theta) = p_{\beta_{k-1}}(\theta)$. Chen et al. [64] note that $p_{\beta_{k-1}}(\theta)$ is an excellent importance distribution because it is slightly less dispersed than $p_{\beta_k}(\theta)$.

The key idea of steppingstone sampling is to express the ratio $Z_1/Z_0$ as the produce of $K$ ratios. By setting, for example $\beta_k = k/K, k = 1, 2, \ldots, K$, the SS estimator relies on noting that:

$$p_{SS}(\mathbf{D}|M) = \frac{Z_1}{Z_0} = \left(\frac{Z_1}{Z_{\beta_{K-1}}}\right)\left(\frac{Z_{\beta_{K-1}}}{Z_{\beta_{K-2}}}\right)\cdots\left(\frac{Z_{\beta_k}}{Z_{\beta_{k-1}}}\right)\left(\frac{Z_{\beta_{k-1}}}{Z_{\beta_{k-2}}}\right)\cdots\left(\frac{Z_{\beta_2}}{Z_{\beta_1}}\right)\left(\frac{Z_{\beta_1}}{Z_0}\right) = \prod_{k=1}^{K} \frac{Z_{\beta_k}}{Z_{\beta_{k-1}}} = \prod_{k=1}^{K} r_k \tag{20}$$

By assuming $g(\theta) = p_{\beta_{k-1}}(\theta)$, the desired ratio $r_k$ can be expressed as:

$$
\begin{aligned}
r_k &= \frac{Z_{\beta_k}}{Z_{\beta_{k-1}}} = \frac{\int q_{\beta_k}(\theta)d\theta}{\int q_{\beta_{k-1}}(\theta)d\theta} = \frac{\int \left(\frac{q_{\beta_k}(\theta)}{q_{\beta_{k-1}}/Z_{\beta_{k-1}}}\right)p_{\beta_{k-1}}(\theta)d\theta}{\int \left(\frac{q_{\beta_{k-1}}}{q_{\beta_{k-1}}/Z_{\beta_{k-1}}}\right)p_{\beta_{k-1}}(\theta)d\theta} \\
&= \int \left(\frac{q_{\beta_k}(\theta)}{q_{\beta_{k-1}}(\theta)}\right)p_{\beta_{k-1}}(\theta)d\theta = \int \left(\frac{p(\mathbf{D}|\theta,M)^{\beta_k}p(\theta|M)}{p(\mathbf{D}|\theta,M)^{\beta_{k-1}}p(\theta|M)}\right)p_{\beta_{k-1}}(\theta)d\theta \\
&= E_{p_{\beta_{k-1}}}\left[\frac{p(\mathbf{D}|\theta,M)^{\beta_k}p(\theta|M)}{p(\mathbf{D}|\theta,M)^{\beta_{k-1}}p(\theta|M)}\right] = E_{p_{\beta_{k-1}}}\left[p(\mathbf{D}|\theta,M)^{\beta_k-\beta_{k-1}}\right] \\
&= \frac{1}{n}\sum_{i=1}^{n} p(\mathbf{D}|\theta_{k-1,i},M)^{\beta_k-\beta_{k-1}}
\end{aligned}
\tag{21}
$$

Accordingly, Equation (19) is obtained by substituting Equation (21) into Equation (20). To improve the numerical stability, we factor out the largest sampled likelihood value $L_{\max,k} = \max_{1<i<n}[p(\mathbf{D}|\theta_{k-1,i},M)]$:

$$
\hat{r}_k = \frac{1}{n}(L_{\max,k})^{\beta_k-\beta_{k-1}}\sum_{i=1}^{n}\left(\frac{p(\mathbf{D}|\theta_{k-1,i},M)}{L_{\max,k}}\right)^{\beta_k-\beta_{k-1}}.
\tag{22}
$$

Substituting Equation (21) in Equation (22) into Equation (20) yields the SS estimator:

$$
\hat{p}_{SS}(\mathbf{D}|M) = \prod_{k=1}^{K}\hat{r}_k = \prod_{k=1}^{K}\frac{1}{n}(L_{\max,k})^{\beta_k-\beta_{k-1}}\sum_{i=1}^{n}\left(\frac{p(\mathbf{D}|\theta_{k-1,i},M)}{L_{\max,k}}\right)^{\beta_k-\beta_{k-1}}.
\tag{23}
$$

Although the derivation of both TI and SS shows that these methods are mathematically rigorous, theoretically unbiased estimators could have a bias in practice arising from numerical errors. For example, if any of the ratios $r_k$ (Equation (22)) suffers from MCMC sampling error this error will propagate through the multiplication as illustrated in Section 4. Similarly, with TI, an error in the mean potential $y_k$ (Equation (13)) will result in the cumulative addition of this error when substituted in Equation (12). Thus, this cumulative property of multiplication (SS) and integration (TI) will result in the systematic overestimation or underestimation of the BME. We further introduce the multiple one-steppingstone sampling (MOSS), which is a variant of SS that is less susceptible to sampling error. MOSS estimates the BME by taking the mean of multiple one-steppingstone $SS_{1,k}$ that steps from the prior to an intermediate distribution $k$ (i.e., ratio $r_{k-1} = r_0$) and from which to the posterior distribution (i.e., ratio $r_K$):

$$
\begin{aligned}
p_{MOSS}(\mathbf{D}|M) &= \frac{Z_1}{Z_0} = \frac{1}{K}\sum_{k=1}^{K}SS_{1,k} = \frac{1}{K}\sum_{k=1}^{K}\frac{Z_{\beta_{k-1}}}{Z_0}\frac{Z_1}{Z_{\beta_{k-1}}} = \frac{1}{K}\sum_{k=1}^{K}r_{k-1}\times r_K \\
&= \frac{1}{K}\sum_{k=1}^{K}\left(\frac{1}{n}\sum_{i=1}^{n}p(\mathbf{D}|\theta_{0,i},M)^{\beta_{k-1}-\beta_0}\right)\left(\frac{1}{n}\sum_{i=1}^{n}p(\mathbf{D}|\theta_{k-1,i},M)^{\beta_K-\beta_{k-1}}\right)
\end{aligned}
\tag{24}
$$

such that $p(\mathbf{D}|\theta_{0,i},M)$ is the likelihood of parameter sample $\theta_{0,i}$ drawn from the prior distribution, and $p(\mathbf{D}|\theta_{k-1,i},M)$ is the likelihood of parameter sample $\theta_{k-1,i}$ drawn from any intermediate distribution that spans from the prior distribution (where $\beta_0 = 0$) to the posterior distribution (where $\beta_K = 1$). Note the special case of $k = 1$ is equivalent to the AM estimator (Equation (17)). Equation (24) does not eliminate the sampling error, but reduces its cumulative effect. The second advantage of Equation (24) is that all the summation terms $\sum_{k=2}^{K}r_{k-1}\times r_K$ repeatedly use the ratio $r_{k-1} = r_0$, which is free from sampling errors that are introduced by MCMC, as $r_0$ can be directly sampled from the prior. However, using only two ratios makes MOSS more susceptible to discretization error, which is an error arising from not using a sufficient number of $\beta$ values. Thus, we expect MOSS to

work for the problem with a low or moderate number of dimensions and not very peaked likelihoods. To improve the numerical stability we use multiple $\hat{r}_k$ (Equation (22)) yielding the MOSS estimator:

$$
\begin{aligned}
\hat{p}_{MOSS}(\mathbf{D}|M) \quad &= \frac{1}{K} \sum_{k=1}^{K} \hat{r}_0 \times \hat{r}_k \\
&= \frac{1}{K} \sum_{k=1}^{K} \left[ \begin{array}{c} \left( \frac{1}{n} (L_{\max,0})^{\beta_{k-1}-\beta_0} \sum_{i=1}^{n} \frac{p(\mathbf{D}|\theta_{0,i},M)^{\beta_{k-1}-\beta_0}}{L_{\max,0}} \right) \\ \times \left( \frac{1}{n} (L_{\max,k})^{\beta_K-\beta_{k-1}} \sum_{i=1}^{n} \frac{p(\mathbf{D}|\theta_{k-1,i},M)^{\beta_K-\beta_{k-1}}}{L_{\max,k}} \right) \end{array} \right]
\end{aligned}
\tag{25}
$$

such that $L_{\max,0}$ and $L_{\max,k}$ are the largest sampled likelihood values from the prior distribution and any intermediate distribution, respectively.

## 3. Gaussian Model Example

We first compare the previously introduced five estimators using a Gaussian model because the true BME is available analytically and we can make direct draws from the power posteriors avoiding errors specific to MCMC sampling. We use a Gaussian model parameterized by $\theta = (\theta_1, \theta_2, \ldots, \theta_D)$ with $D$ dimensions. Following [22], we define the likelihood as a product of:

$$
p(M|\theta) = \prod_{d=1}^{D} e^{-\frac{\theta_d^2}{2v}}
\tag{26}
$$

where $v = 1$ is a hyperparameter. The prior is the product of independent normal distributions of each parameter $\theta_d$ sampled from a standard normal distribution. Samples for all power posterior distributions can be directly drawn from a normal distribution:

$$
\theta_{d,\beta_k} \sim N\left(0, \sqrt{\frac{v}{\beta_k + v}}\right)
\tag{27}
$$

with mean 0 and variance $v/(\beta_k + v)$ such that the prior(i.e., $\beta_0 = 0$) is the product of $D$ independent standard normal distributions of each $\theta_{d,0}$; the posterior (i.e., $\beta_K = 1$) is the product of $D$ independent distributions of each $\theta_{d,1}$ with mean 0 and variance $v/(1 + v)$; and the true analytical solution (AS) of the BME is

$$
p(M) = \sqrt[D]{\frac{v}{1 + v}}
\tag{28}
$$

for a model with $D$ dimensions.

Using Equation (14) with $\alpha = 0.3$, we parameterize a path of $\beta_k$, $k = 0, 1, \ldots, K$ from $\beta_0 = 0$ to $\beta_K = 1$ with $K + 1$ steps, and with the sample size $n = 10,000$ for each step $\beta_k$. Since we can use the same $\beta$ discretization for TI, SS, and MOSS. Generally, we can always make the most of our samples by evaluating the BME using these three estimators without any extra computational effort. The only exception is that SS estimators (Equations (23) and (25)) do not need to sample the $K + 1$ power posterior distribution (i.e., the posterior distribution $p_1$). Thus in this estimator comparison exercise, the SS estimators are slightly disadvantaged as they need only $N = n(K)$ samples, while AM, HM, and TI estimator needs $N = n(K + 1)$ samples. Note that for AM and HM the tuning parameter $\alpha$ is irrelevant and $K$ merely indicates sample size $N = n(K + 1)$ from the prior and posterior, respectively. We report the mean BME and its relative error based on 10 independent estimations. The relative error is defined as

$$
\delta = \frac{p_{est}(.) - p_{ref}(.)}{p_{ref}(.)}
\tag{29}
$$

where $p_{ref}(.)$ and $p_{est}(.)$ are the reference solution and estimated solution, respectively. For this Gaussian model, the reference solution $p_{ref}(.)$ is a true analytical solution (Equation (28)). We compare

different estimators based on accuracy, where better accuracy is defined as less bias from the reference solution given an equal number of sample draws.

We start by observing the effect of dimensionality (D = 1, 50, and 100) on different estimators. Table 1 shows that all the estimators work well for the simple one-dimensional case resulting in a relative error less than ±0.01%. When increasing the problem dimensionality, the performance of different estimators diverged. AM tends to underestimate the BME as the number of dimensions increases with relative error decreasing from −3.18% for the 50-dimensional problem to −30.8% for the 100-dimensional problem. HM avoided underestimating through enriching the BME estimation with samples from high-likelihood regions. However, this resulted in overestimation, which makes the model appear better-fitting than it actually is. One might argue that this HM comes at no extra computational cost as we need to sample the posterior anyway, and such overestimation is irrelevant as we are not generally interested in the absolute value of the BME, but in the relative model weight of candidate models (Equation (3)). This is true as long as the estimated BME covariates the true BME. However, the results indicate that the magnitude of overestimation increases with increasing the model complexity. Overestimation becomes much larger as the number of dimension increases with relative error increasing from 20.72% for the 50-dimensional problem to 189.34% for the 100-dimensional problem. This is mainly because HM is less sensitive to the prior, and thus inaccurately accounts for the model complexity. That can be explained by looking at the reference BME solution (Table 1), which decreased from $2.98 \times 10^{-8}$ for the 50-dimensional to $8.88 \times 10^{-16}$ for the 100-dimensional problem due to the presence of large parameter space with low-likelihood. Increasing the contribution of low-likelihood regions is equivalent to decreasing the informativeness of prior (either due to increasing the number of dimensions or using wider parameter space for each dimension). HM tends to ignore low-likelihood regions in the parameter space, which means that it will impose less plenty than it should on complex model with superfluous parameters. This can impact model ranking as shown in the groundwater transport model selection problem in the next section. On the other hand, the performance of TI, SS, and MOSS for the 50- and 100-dimensional problems is relatively stable with relative error of less than 1%. However, this comparison is based on a fine discretization of the sampling path with $K = 50$, which is not computationally efficient.

**Table 1.** True analytical solution (AS) of the BME for the Gaussian model with different dimensions (D), and the corresponding numerical estimations with K = 50 and $\alpha$ = 0.3 using arithmetic mean (AM), harmonic mean (HM), thermodynamic integration (TI), steppingstone sampling (SS), and multiple one-steppingstone sampling (MOSS). The mean BME and its relative error ($\delta$) are based on 10 independent estimations.

|  | D = 1 | | D = 50 | | D = 100 | |
|---|---|---|---|---|---|---|
|  | **Mean** | **$\delta$** | **Mean** | **$\delta$** | **Mean** | **$\delta$** |
| AS | 0.707107 | - | 2.98E−08 | - | 8.88E−16 | - |
| AM | 0.706716 | −0.06% | 2.89E−08 | −3.18% | 6.15E−16 | −30.80% |
| HM | 0.70718 | 0.07% | 3.6E−08 | 20.72% | 2.57E−15 | 189.34% |
| TI | 0.707131 | −0.01% | 2.98E−08 | −0.17% | 8.85E−16 | −0.32% |
| SS | 0.707143 | 0.01% | 2.98E−08 | 0.02% | 8.89E−16 | 0.04% |
| MOSS | 0.707147 | 0.01% | 2.99E−08 | 0.24% | 8.8E−16 | −0.97% |

The number of steps $\beta_k$ and their distribution according to the parameter $\alpha$ are important for TI, SS, and MOSS. One might wonder if $K = 50$ is large enough for this example, and to what extent using coarser discretization would impact these estimators. For the 100-dimensional example and setting $\alpha = 0.3$, we make the $\beta_k$ steps coarser using $K = 10$ and $K = 5$, respectively. Table 2 indicates that very coarse discretization $K = 5$ is sufficient for SS and MOSS estimator yielding a relative error of less than 1%, while TI relative error for $K = 10$ is −8.21% and deteriorats to −29.06% for $K = 5$. It is

worth noting the AM and HM estimates deteriorated as well due to using fewer numbers of prior and posterior samples, respectively.

**Table 2.** BME estimation for the Gaussian model with D = 100 dimensions and $\alpha$ = 0.3 using the arithmetic mean (AM), harmonic mean (HM), thermodynamic integration (TI), steppingstone sampling (SS), and multiple one-steppingstone sampling (MOSS). The mean BME and its relative error ($\delta$) are based on 10 independent estimations. The true analytical solution is 8.88E−16.

|  | K = 5 |  | K = 10 |  | K = 50 |  |
|---|---|---|---|---|---|---|
|  | **Mean** | **$\delta$** | **Mean** | **$\delta$** | **Mean** | **$\delta$** |
| AM | 5.13E−16 | −42.27% | 5.21E−16 | −41.31% | 6.15E−16 | −30.80% |
| HM | 4.44E−15 | 399.94% | 4.19E−15 | 371.42% | 2.57E−15 | 189.34% |
| TI | 6.3E−16 | −29.06% | 8.15E−16 | −8.21% | 8.85E−16 | −0.32% |
| SS | 8.95E−16 | 0.72% | 8.89E−16 | 0.08% | 8.89E−16 | 0.04% |
| MOSS | 8.95E−16 | 0.78% | 8.86E−16 | −0.23% | 8.8E−16 | −0.97% |

Additionally, a good estimator should have fewer parameters to tune. Using the same 100-dimensional example, we examine to what extent TI, SS, and MOSS are sensitive to locations of path steps, by fixing $K = 5$ and choosing different values of $\alpha$ as shown in Figure 1. The results indicate that SS and MOSS have the favorable feature of being relatively invariant to $\beta_k$ values with relative error less than −10%. TI considerably depends on locations of the path steps with relative error ranging from about −10% to −80% for different $\beta_k$ values.

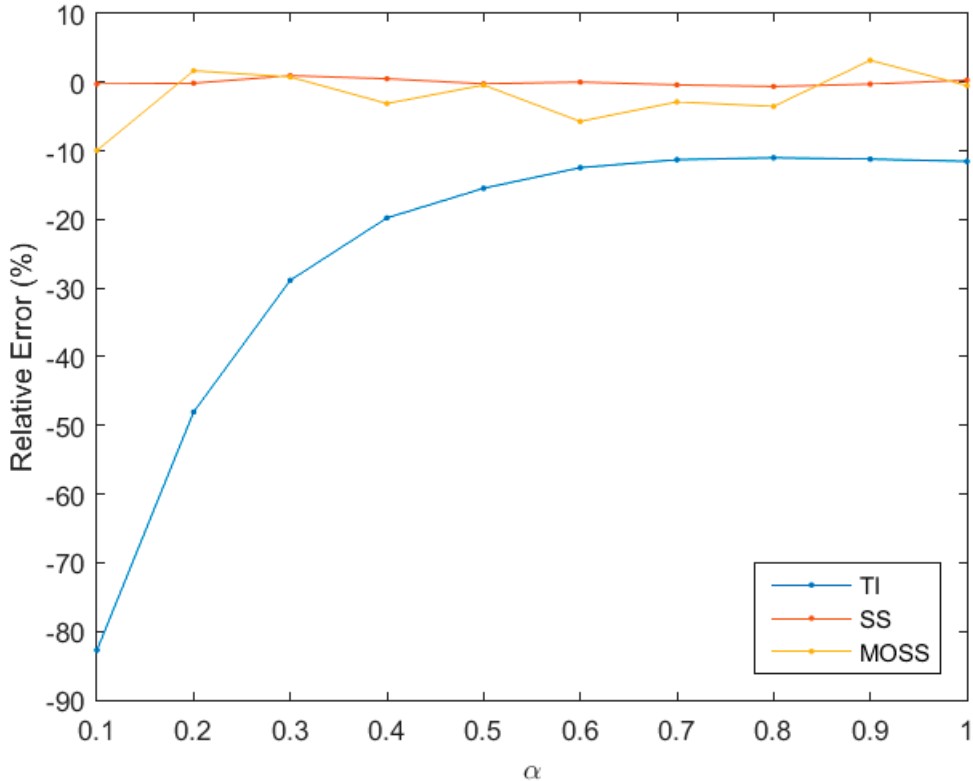

**Figure 1.** Bayesian model evidence (BME) estimation for the Gaussian model with D = 100 dimensions and K = 5 using thermodynamic integration (TI), steppingstone sampling (SS), and multiple one-steppingstone sampling (MOSS). The relative error (%) of the mean BME is based on 10 independent estimations.

This comparison indicates that SS has relatively better performance followed by MOSS and TI for this Gaussian example. However, for this example, parameter samples can be directly sampled from a series of power posterior distributions without MCMC. Yet for almost all practical applications MCMC is needed to approximate the power posterior distributions introducing an MCMC sampling error. In the next section, we examine to what extent this MCMC sampling error can affect the performance of these three estimators. In addition, we examine to what extent the HM under-penalization of model complexity will affect the model ranking.

## 4. Groundwater Transport Models with Different Complexity

### 4.1. Problem Statement

Column experiments and tracer tests are widely used to study the fate and transport of soil and groundwater contaminants by fitting breakthrough curves to transport models. Breakthrough curves are curves of concentrations measured at an observation point downstream from a source plotted against time. Least square regression methods are generally used for fitting breakthrough curves of column experiments and tracer tests with subsurface transport models [84–96]. Subsequently, model weight calculation and ranking of multiple candidate models are carried out using an approximate semi-analytical solution [85,87], which indirectly accounts for model complexity as previously discussed. We examined the use of numerical estimators to estimate the BME of multiple candidate groundwater transport models with evolving model complexity. The models fit a breakthrough curve of a column experiment. The objective is to rank these models after accounting for model complexity through numerical BME estimation.

We selected the data of the miscible displacement column experiment with titrated water from Anamosa et al. [84]. The objective of that column experiment was to determine if preferential water flow and immobile-water regions should be considered to describe solute movement in tropical stone-line soil when developing nutrient management programs. Undisrupted soil column was taken from the top three horizons of clayey-skeletal, oxidic isohyperthermic, Typic Gibbsiorthox soil. We fit the breakthrough curve of Experiment I-3 of Anamosa et al. [84] using four groundwater transport models with evolving complexity and two different model structures. The first model structure is the traditional equilibrium advection-dispersion equation (ADE) model with linear adsorption [97]:

$$R\frac{\partial C}{\partial T} = -\frac{\partial C}{\partial X} + \frac{1}{P}\frac{\partial^2 C}{\partial X^2} \tag{30}$$

with retardation factor $R = 1 + \rho_s k_d / \theta$ where $\theta = 0.525$ is the volumetric water content [$L^3 L^{-3}$ or $L^\circ$], $\rho_s$ [$ML^{-3}$] is the soil bulk density; and $k_d$ is a sorption partition coefficient [$L^3 M^{-1}$]. For consistency with the original analysis we fix the retardation factor $R$ at 1.05. $C = c/c_0$ is the normalized solute concentration [-], where $c$ and $c_0$ are the effluent and influent tracer concentrations [$ML^{-3}$], respectively. The dimensionless pulse duration $T = qt/\theta L = vt/L$ is solute pulse volume [-] such that $q = 2.71$ cm d$^{-1}$ is the Darcy flux [$LT^{-1}$], $v$ is the mean pore water velocity [$LT^{-1}$], $t$ is time [T] and $L = 71.6$ cm is the column length [L]. $X = x/L$ is the relative distance where $x$ is the distance from the inlet boundary. $P = vL/D = L/\lambda$ is the Peclet number [-], where $D$ is the dispersion coefficient [$L^2 T^{-1}$] and $\lambda$ is the dispersivity [L]. The initial condition is $c(x,0) = 0$. The inlet and outlet boundary conditions are described by [97]. At the inlet boundary, a pulse input of concentration $c_0$ is applied from $t = 0 - t_0$ with pulse duration $t_0 = 39.39$ days. Note that the dimensionless pulse duration $T_0$ is 2.84 pore volumes.

We define ADE1 as the first and simplest model in which dispersivity $\lambda$ is the only calibrated parameter with a uniform prior 0.001 to 200 cm. Anamosa et al. [84] estimated that stagnant or immobile water regions were about 50% of the total water content. In the presence of diffusive mass transfer of water into stagnant or immobile regions, the equilibrium ADE model is often unable to simulate the breakthrough curves. To make up for this model structure error, ADE2 model additionally varies water content $\theta$ since the equilibrium ADE model is most sensitive to this parameter [86] .

The measured volumetric water content is $\theta = 0.525$ [84], and ADE2 estimates this parameter using a uniform prior from 0.1 to 0.7.

Since the inadequacy of the equilibrium ADE model to simulate breakthrough curves is generally attributed to the presence of non-equilibrium processes, we simulate the breakthrough curve using a physical non-equilibrium model. We use the two-region mobile-immobile model (MIM) [97]:

$$\beta R \frac{\partial C_1}{\partial T} = -\frac{\partial C_1}{\partial X} + \frac{1}{P} \frac{\partial^2 C_1}{\partial X^2} - \omega(C_1 - C_2) \tag{31}$$

and

$$(1 - \beta)R \frac{\partial C_2}{\partial T} = \omega(C_1 - C_2) \tag{32}$$

The mobile-immobile water partition coefficient $\beta = (\theta_m + f\rho_s K_d)/(\theta + \rho_s K_d)$ [-] is a fraction of the solute present in the mobile region under equilibrium conditions, in which $\theta_m$ is the mobile pore water content and $f$ is the fraction of the total sorption sites that equilibrate with the solute in the mobile region; $C_1$ and $C_2$ are normalized concentrations (normalized with respect to $c_0$) [-] in the mobile and immobile regions, respectively; $\omega = \alpha L/(\theta v)$ is a dimensionless mass transfer coefficient between mobile-immobile water [-] with first-order mass transfer coefficient $\alpha$ [T$^{-1}$] . Note that for $\beta = 1$ and $\omega = 0$, the physical non-equilibrium MIM model in Equation (31) reduces to the equilibrium ADE model in Equation (30). The analytical solutions for both ADE and MIM models are given in [97] and model simulations are implemented using CXTFIT/Excel [86,98]

MIM is a high fidelity model as it can better describe asymmetric breakthrough curves of titrated water for this two-region soil type by accounting for non-equilibrium transport [84,86]. Fidelity refers to the realism of representing our scientific knowledge about a real-world system. We consider two alterative MIM models, MIM1 and MIM2. The most simple high fidelity MIM1 model that has three parameters. Dispersivity $\lambda$ is estimated similar to the two ADE models with the same prior distribution. The two additional parameters are mobile-immobile water partition coefficient $\beta$ with a uniform prior from 0.1 to 0.999, and the dimensionless mass transfer coefficient $\omega$ [-] between mobile-immobile water with a uniform prior from 0.01 to 100. As mass balance error could impact parameter estimation [97,99]; MIM2 additionally estimates the pulse duration $T_0$ with a uniform prior distribution from 2.56 to 3.12. This range is based on assuming a ±10% error from measured dimensionless pulse duration $T_0 = 2.84$. Although maintaining a constant injection rate for about 40 days is not a trivial task, assigning a ±10% mass balance error appears as an unnecessary complexity.

Note that [84] conducted several experiments with different Darcy flux $q$. For example, take the two cases of $q = 2.71$ cm d$^{-1}$ and $q = 111$ cm d$^{-1}$, respectively. The high Darcy flux $q = 111$ cm d$^{-1}$ is a special case since equilibrium is not reached at high velocity, and thus the performance of ADE and MIM models will be very similar. At low velocity, the performance of ADE models will deteriorate as it does not account for non-equilibrium transport. Thus, we selected the experiment with $q = 2.71$ cm d$^{-1}$ that represents the general case. Also, note that the fixed input model parameters are experimentally measured

### 4.2. Reference Solution

For model calibration we used standard least square (SLS) likelihood function with a standard error equal to 10% from the mean observation, which is equivalent to a variance of $1.8 \times 10^{-3}$. Figure 2 shows that the likelihood space of the posterior is very peaked relative to the prior such that the average of log-likelihood $y_k$ (Equation (13)) varies from −462.2 at the prior to 60.9 at the posterior for ADE2 model. Unlike the previous problem discussed in Section 3, there is no true analytical solution of BMA as all the power posterior distributions are non-Gaussian as shown in Figure 2. To facilitate the comparison of different estimators we estimated the reference solutions of the four candidate models by sampling the prior using 5 million samples for each model and calculating the BME using an AM estimator (Equation (17)).

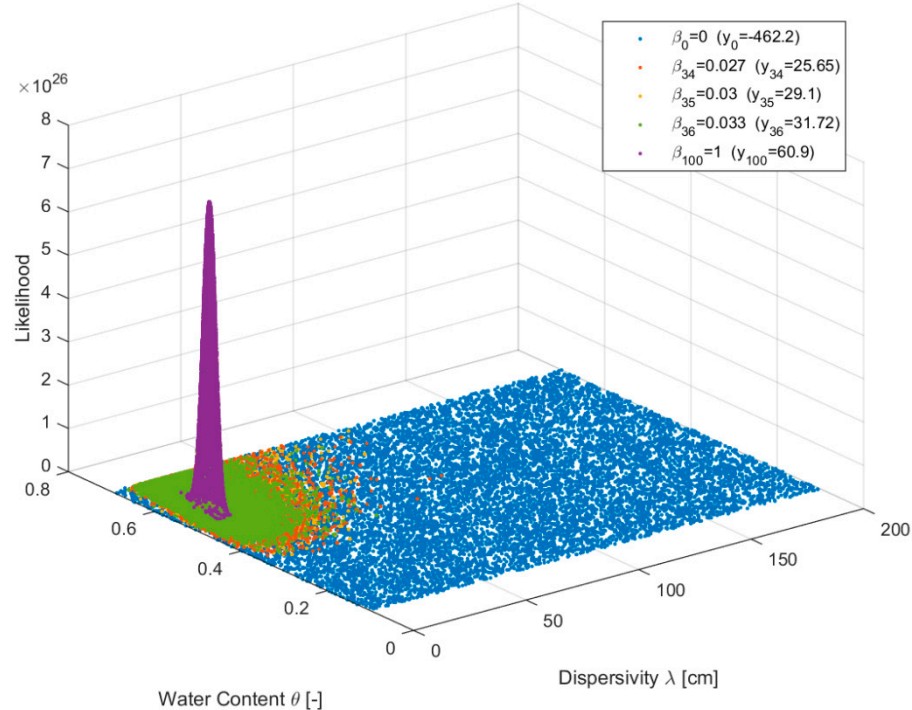

**Figure 2.** Likelihood surface at different $\beta_k$ values of model ADE2. The number of power posterior coefficients $\beta_k$ are $K+1$ with $K = 100$, and their values are determined by Equation (14) using the shape parameter $\alpha = 0.3$. We also report the potential expectation $y_k$ (Equation (13)) for each distribution.

Model ranking using simple metrics such as root mean squared error (RMSE) that does not account for model complexity is insufficient, and BME estimation is important because it naturally accounts for model complexity. As expected, normalized RMSE (i.e., normalized by mean observation) of the best realization is the highest 5.4% for the simplest model ADE1 and the lowest 3.737% for model MIM2 as shown in Table 3 for a realization with the best model fit to data. However, penalizing model complexity using BME, we see that the most complex model MIM2 with the best fitting was penalized and became the least plausible model, and that the simplest model ADE1 is not necessarily the best model. By accounting for model complexity, the high fidelity model MIM1 has the highest BME with a model weight of 50.92% as shown in Table 3, in which model weights are calculated using Equation (3) assuming equal prior model probabilities. Despite its higher complexity, it is not surprising that the high-fidelity MIM1 model will stand out. We already know that MIM models represent true physics as the ratio of immobile water is at least 0.47 (Anamosa et al., 1990). However, there is a trade-off between model fidelity and model complexity. The most complex high-fidelity MIM2 model has the lowest BME with a model weight of 8.06%, as the unnecessary complexity was highly penalized. This is not the case for the ADE model. The ADE2 model has higher BME (i.e., model weight of 22.66%) than the ADE1 model (i.e., model weight of 18.36%), as the ADE2 model made up for the model structure error through calibrating an extra parameter. However, although MIM1 is more complex than ADE2, MIM1 has the highest BME as its extra complexity is justified.

Here we merely want to show how extra complexity is penalized through BME estimation. Whether this makes sense from a physical point of view with respect to model fidelity or whether this is favorable for the modeling objective, it is a different question since the BME depends on the quality of data, likelihood function, and the prior selection. Here we are more interested in answering the question of how complex models are penalized, and how the accuracy of BME estimation affects the penalization of complex models and accordingly model ranking. For this purpose, we use the AM estimates as the reference solution, which is subject to Monte Carlo stochastic error. The law of large

numbers ensures that the estimate converges to the correct value as the number of samples increases. Monte Carlo stochastic error is given by

$$SE_k = \frac{s_k}{\sqrt{n}} \tag{33}$$

in which $s_k$ is the standard deviation of the likelihoods of any $k$ distributionand $n$ is the number of used samples, such that as $n$ goes to infinite, and the error $SE_k$ goes to zero. For the candidate models, Table 1 shows that the Monte Carlo stochastic error relative to BME is the lowest ±0.29% for model ADE1 and the highest for the ±0.96% model ADE2. The 95% credible interval error of the reference is $1.645 \times SE$, and accordingly, the 95% credible interval of the reference solution is the AM ± $1.645 \times SE$. Given a 95% credible interval, the lowest and the highest relative errors of the reference solution are ±0.48% for model ADE1 and ±1.54% for model MIM2, respectively, as shown in Table 1.

**Table 3.** Metrics and model ranking of the candidate models based on the best realization (i.e., RMSE) and based on the average model fit of all the parameter values that the model can take (i.e., BME and model weight). The results are based on the implementation of AM.

|  | ADE1 | ADE2 | MIM1 | MIM2 |
|---|---|---|---|---|
| RMSE of the best realization | 0.0227 | 0.0121 | 0.0115 | 0.0115 |
| RMSE normalized by mean observation | 5.4% | 2.88% | 2.738% | 2.737% |
| Model rank by the best realization | 4 | 3 | 2 | 1 |
| BME | 6.23E+23 | 7.68E+23 | 1.73E+24 | 2.73E+23 |
| Model weight | 0.1836 | 0.2266 | 0.5092 | 0.0806 |
| Model rank by model weight | 3 | 2 | 1 | 4 |
| BME estimation error | ±1.81E+21 | ±7.34E+21 | ±8.44E+21 | ±2.56E+21 |
| BME estimation error [%] | ±0.29% | ±0.96% | ±0.49% | ±0.94% |
| BME 95% credible interval error [%] | ±0.48% | ±1.57% | ±0.80% | ±1.54% |

### 4.3. Sampling Error

Before comparing the four numerical methods for BME estimation (HM, TI, SS, and MOSS), we first show that theoretically unbiased estimators such as SS and TI could have a large bias in practice arising from numerical implementation such that MCMC sampling errors can introduce bias. In addition, we show how MOSS is less susceptible to sampling error. Using model ADE2, we calculate the relative errors of BME estimates based on a sequence of $\beta_k$ values using SS with $\boldsymbol{\beta} = \{0 = \beta_0, \beta_1, \ldots, \beta_{k-1}, \beta_k, \beta_{k+1}, \ldots, \beta_K = 1\}$. Take for example the case of $K = 100$ and $\alpha = 0.3$. As shown in Figure 3, at $\beta_1 = 2.15 \times 10^{-7}$ the BME estimate is based on $\boldsymbol{\beta} = \{0, \beta_1, 1\}$; at $\beta_2 = 2.17 \times 10^{-6}$ the BME estimate is based on $\boldsymbol{\beta} = \{0, \beta_1, \beta_2, 1\}$ and so on; at the last step $\beta_{100} = 1$ we obtain the regular SS estimate $\boldsymbol{\beta} = \{0, \beta_1, \ldots, 1\}$. Note that $\alpha = 0.3$ is the default case for all following examples. We draw $n = 20,000$ samples at each step using the state of the art MCMC code of MT-DREMA$_{(ZS)}$ [60], and burn-in the first 5000 samples. This burn-in period is the default case for all the following examples. Figure 3a shows that the relative error of the SS-based BME fluctuates with both under and overestimation, and at a certain $\beta_k$ value the BME estimate drops. By zooming in using the log-scale (Figure 3b) this drop is around $\beta_{30} = 0.018$. Although adding the next $\beta_{31} = 0.02$ raised the relative error from about −37% to −23%, the error propagated such that the relative error of the last step $\beta_{100} = 1$ (i.e., the SS estimate) is about −33.23 %. Similarly, errors in mean potential $y_k$ of TI (Equation (13)) resulting from MCMC sampling errors will accumulate through the integration (Equation (12)). Yet as the integration is not series of multiplications as SS (Equation (23)), TI underestimated the BME by −20.62%.

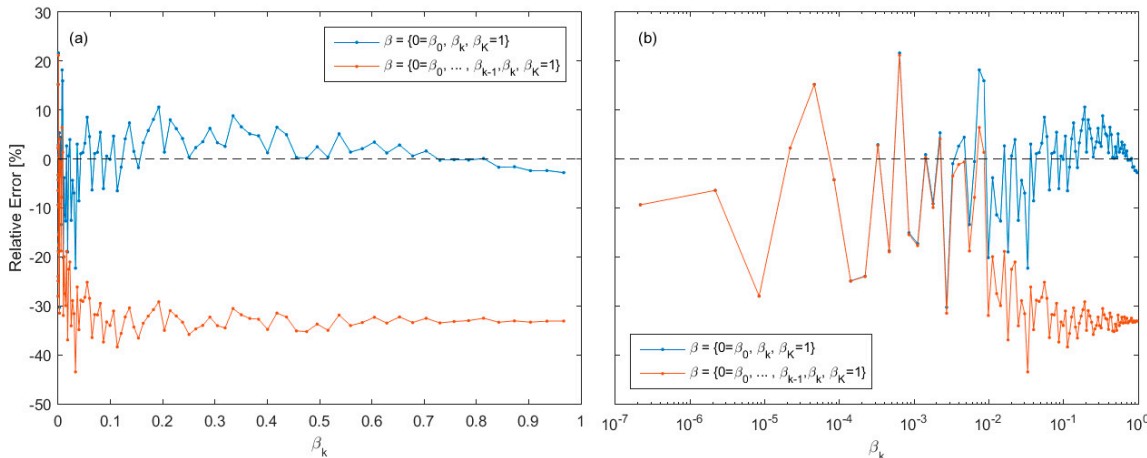

**Figure 3.** The relative error of sequence of BME estimates for $\beta_k$ values expressed in (**a**) standard linear scale and (**b**) log scale based on one-steppingstone sampling with $\boldsymbol{\beta} = \{\beta_0 = 0, \beta_k, \beta_K = 1\}$ and a steppingstone sampling with $\boldsymbol{\beta} = \{\beta_0 = 0, \beta_1, \dots, \beta_k, \beta_K = 1\}$, given K = 100 and $\alpha = 0.3$. Note that the mean of values presented by the blue line is the MOSS estimate, and the last estimated value of the orange line with $\boldsymbol{\beta} = \{\beta_0, \beta_1, \dots, \beta_K\}$ is the SS estimate.

Figure 3 shows that the using one-steppingstone sampling such that $\boldsymbol{\beta} = \{\beta_0 = 0, \beta_k, \beta_K = 1\}$ can alleviate the error accumulation. This is expected given the two reasons that were previously discussed with Equation (24). Note that the relative error of the MOSS estimate, which is the mean of the relative errors of the one-steppingstone sampling sequence shown in Figure 3, is equal to 1.1%. In other words, the "blocks" that make the SS stumbles $\prod_{k=1}^{K} r_k = \prod_{k=1}^{K} \frac{Z_{\beta_k}}{Z_{\beta_{k-1}}}$, the MOSS turns them into steppingstones toward the BME estimation through averaging one-steppingstones $\frac{1}{K} \sum_{k=1}^{K} r_{k-1} \times r_k = \frac{1}{K} \sum_{k=1}^{K} \frac{Z_{\beta_{k-1}}}{Z_0} \frac{Z_1}{Z_{\beta_{k-1}}}$. This is based on the relative advantage that steppingstones sampling methods require less discretization. Although the MOSS estimator might not hold for more complex sampling spaces, here were are mainly interested in showing the concept that MCMC sampling error can be alleviated.

There are two sources of error, which are the sampling error and discretization error. There are at least three sources of sampling errors. First, one needs to allow the chain to acclimate in the target distribution by using an adequate burn-in period and enough sampling. This is to satisfy the stationary assumption that the mean, variance, and autocorrelation structure do not change over time. Second are errors from MCMCs inadequate approximation of the target distribution with increasing level of sampling complexity. Third are errors arising from numerical instability due to a rounding error for example. Discretization error is a function of the number and locations the discretization coefficients $\beta_k$ along the path and can be reduced by decreasing the Kullback–Liebler distance between successive intermediate distributions.

While discretization error can be resolved by using more intervals, the sampling error might not ultimately be eliminated through more sampling efforts. Given the total number of samples $n \times K = 2,000,000$, Table 4 shows that TI is more sensitive to discretization errors, while SS is more sensitive to sampling errors. Such that for the small number of intervals K = 5 and the accordingly larger number of samples per interval $n = 400,000$, the TI has a relative error of −98.23%, while SS has a relative error of −16.67%. On the other hand, for large K = 100 and accordingly smaller $n = 20,000$, the relative errors of TI and SS are −20.62% and −33.23%, respectively. The relative error of MOSS for different $K \in \{5, 10, 20, 100, 200\}$ is much smaller ranging from −3.27% to 1.17%. Note that given a 95% credible interval, the reference solution error of model ADE2 is ±1.57%. Table 5 confirms that SS is more sensitive to sampling errors. In comparison to Table 4, by fixing $n = 20,000$ and varying $K \in \{5, 10, 20, 100\}$ the SS estimates show a larger relative error when using smaller sample

size $n$, while the TI is relatively invariant. On the other hand, increasing $K$ intervals reduces the TI bias asymptotically. Note that TI bias cannot be reduced beyond a certain limit due to the presence of sampling error, which cannot be eliminated by finer discretization. MOSS is less sensitive to the sample size $n$, showing a relative error from −3.91% to 1.17% for different $K$ intervals when $n = 20,000$. Generally, Tables 4 and 5 show that increasing the number of samples $n$ can reduce the sampling error when using SS. The best results will be obtained for a relatively small $K$ with a larger sample size $n$ as shown in Table 6. Using small $K = 5$ intervals and increasing the sample size $n$, SS shows a consistent reduction in relative errors from −25.52% to −16.67% for $n = 10,000$ and $n = 400,000$ samples, respectively. Given the same $K$, increasing the $n$ number of samples reduces the SS bias up to a certain limit. Thus, in theory, finer discretization of the sampling path through using more $K$ intervals should reduce TI and SS estimation bias, yet this is not always the case in practice due to numerical difficulties.

**Table 4.** BME (BME Relative Error [%]) for ADE2 model using thermodynamic integration (TI), steppingstone sampling (SS), and multiple one-steppingstone sampling (MOSS), for varying sample size $n$ per $K$ interval such that $n \times K = 2,000,000$. The reference BME solution is $7.68 \times 10^{23} \pm 7.34 \times 10^{21}$, and the 95% credible interval of the reference solution error is ±1.57%.

| K | TI | SS | MOSS |
|---|---|---|---|
| 5 | 1.38E+22 (−98.20%) | 6.40E+23 (−16.67%) | 7.48E+23 (−2.60%) |
| 10 | 2.71E+23 (−64.71%) | 6.02E+23 (−21.61%) | 7.44E+23 (−3.13%) |
| 20 | 4.52E+23 (−41.15%) | 5.58E+23 (−27.34%) | 7.51E+23 (−2.21%) |
| 100 | 6.10E+23 (−20.57%) | 5.13E+23 (−33.20%) | 7.77E+23 (1.17%) |
| 200 | 6.01E+23 (−21.76%) | 4.93E+23 (−35.87) | 7.617E+23 (−3.27%) |

**Table 5.** BME (BME Relative Error [%]) for ADE2 model using thermodynamic integration (TI), steppingstone sampling (SS), and multiple one-steppingstone sampling (MOSS), for $n = 20,000$ number of samples per $K$ interval. The reference BME solution is $7.68 \times 10^{23} \pm 7.34 \times 10^{21}$, and the 95% credible interval of the reference solution error is ±1.57%.

| K | TI | SS | MOSS |
|---|---|---|---|
| 5 | 1.24E+22 (−98.39%) | 5.76E+23 (−25.00%) | 7.38E+23 (−3.91%) |
| 10 | 2.65E+23 (−65.49%) | 5.99E+23 (−22.01%) | 7.41E+23 (−3.52%) |
| 20 | 4.50E+23 (−41.41%) | 5.00E+23 (−34.90%) | 7.50E+23 (−2.34%) |
| 100 | 6.10E+23 (−20.57%) | 5.13E+23 (−33.20%) | 7.77E+23 (1.17%) |

**Table 6.** BME (BME Relative Error [%]) for ADE2 model using thermodynamic integration (TI), steppingstone sampling (SS), and multiple one-steppingstone sampling (MOSS), for $K = 5$ with a varying $n$ number of samples per $K$ interval. The reference BME solution is $7.68 \times 10^{23} \pm 7.34 \times 10^{21}$, and the 95% credible interval of the reference solution error is $\pm 1.57\%$.

| n | TI | SS | MOSS |
|---|---|---|---|
| 1000 | 1.23E+22 (−98.40%) | 5.72E+23 (−25.52%) | 7.44E+23 (−3.13%) |
| 100,000 | 1.29E+22 (−98.32%) | 6.27E+23 (−18.36%) | 7.57E+23 (−1.43%) |
| 200,000 | 1.29E+22 (−98.32%) | 6.33E+23 (−17.58%) | 7.47E+23 (−2.73%) |
| 300,000 | 1.33E+22 (−98.27%) | 6.35E+23 (−17.32%) | 7.46E+23 (−2.86%) |
| 400,000 | 1.38E+22 (−98.20%) | 6.40E+23 (−16.67%) | 7.48E+23 (−2.60%) |

*4.4. Penalizing Model Complexity*

In this section, we investigate the effect of the accuracy of the BME estimation on penalizing model complexity when using HM, TI, SS, and MOSS estimators. Since TI, SS, and MOSS are more computationally expensive in comparison to HM, we are also interested in how SS and MOSS can save on the computational cost to estimate the BME, and how penalizing model complexity will be impacted when using fewer $\beta$ values. We estimated the BME of the four ADE and MIM models that were previously defined using $K = 5$ with $n = 10,000$ samples per $K$ interval.

The objective is to rank the candidate models after accounting for model complexity. The model ranking according to the reference solutions is MIM1, ADE2, ADE1, and MIM2. Table 7 shows that HM overestimated the BME. This overestimation becomes more pronounced when adding complexity, given the same model fidelity. For example, the relative errors of HM estimates for AME1 and AME2 models are 894% and 5121%, respectively. Similarly, the relative errors of HM estimate increased from 1168% for MIM1 model to 2834% for MIM2 model. This alters the model ranking such that more complex models are generally favored, and thus AME1 receives the lowest model weight. On the other hand, as important corners in the TI integral are missing due to using fewer $\beta$ values, TI underestimated the BME such that relative errors of the four candidate model ranged from −93.35% to −98.4%. We also notice that given the same model fidelity, TI underestimation slightly increases when adding more complexity. This resulted in changing the model ranking with respect to the reference solution such that simple models are more favored, given the same model fidelity. For example, ADE1 receives larger model weight than ADE2, and thus changing the ranking with respect to the reference solution.

Alternatively, SS and MOSS yield the same model ranking with respect to a reference solution. SS also underestimated the BME, yet the model ranking did not change. This is mainly because the relative errors of the four candidate models have a large range from −10.53% to −44.54% for SS as compared to −3.35% to −98.4% for TI. This indicates that SS seems to better covariate with the reference values. For example, given the same model fidelity, SS underestimation considerably increases when adding more complexity. MOSS gives the most accurate results with relative errors for the candidate models ranging from −3.37% to 4.45%. Although the BME values are slightly outside the 95% credible intervals of the relative errors of the reference solutions, MOSS still has a smaller bias compared to HM, TI, and SS.

**Table 7.** BME (BME Relative Error [%]) and model weight (model ranking) for the candidate modes using harmonic mean (HM), thermodynamic integration (TI), steppingstone sampling (SS) and multiple one-steppingstone sampling (MOSS), for $K = 5$ and $n = 10,000$ samples per $K$ interval.

|  | AME1 | | AME2 | | MIM1 | | MIM2 | |
|---|---|---|---|---|---|---|---|---|
|  | BME (Error) | Weight (Ranking) | BME (Error) | Weight (Ranking) | BME (Error) | Weight (Ranking) | BME (Error) | Weight (Ranking) |
| Reference | 6.23E+23 (±0.29%) | 0.1836 (3) | 7.68E+23 (±0.96%) | 0.2263 (2) | 1.73E+24 (±0.49%) | 0.5097 (1) | 2.73E+23 (±0.94%) | 0.0804 (4) |
| HM | 6.20E+24 (894.43%) | 0.0812 (1) | 4.01E+25 (5121.43%) | 0.53 (1) | 2.19E+25 (1168.72%) | 0.2878 (2) | 8.01E+24 (2834.03%) | 0.11 (3) |
| TI | 1.61E+22 (−97.41%) | 0.11 (2) | 1.23E+22 (−98.40%) | 0.0824 (3) | 1.12E+23 (−93.53%) | 0.7507 (1) | 8.78E+21 (−96.78%) | 0.0589 (4) |
| SS | 5.57E+23 (−10.53%) | 0.2096 (3) | 5.72E+23 (−25.52%) | 0.2151 (2) | 1.38E+24 (−20.33%) | 0.5183 (1) | 1.51E+23 (−44.52%) | 0.0570 (4) |
| MOSS | 6.02E+23 (−3.37%) | 0.1794 (3) | 7.44E+23 (−3.13%) | 0.2217 (2) | 1.72E+24 (−0.36%) | 0.5138 (1) | 2.85E+23 (4.45%) | 0.0850 (4) |

## 5. Conclusions

Although the comparison of semi-analytical solutions for estimating the BME has been carried out before [38] as well as the comparison of semi-analytical solutions with Monte Carlo simulation methods [21,24,26], to our knowledge this is the first study in hydrology that examines, in detail, the path sampling and importance sampling Monte Carlo numerical methods for estimating the BME. This study introduces the importance sampling methods of SS and MOSS for BME estimation. By comparing these two estimators with the path sampling method of TI, this study shows that SS is more computationally efficient, requiring fewer path steps and relatively invariant to the steps location. Theoretically, both TI and SS are unbiased estimators, yet a theoretically unbiased estimator could have a large bias in practice arising from the numerical implementation such that MCMC sampling errors can introduce bias. The study shows that MOSS can reduce the impact of sampling errors, yet this method is mainly limited for moderately complex problems where fewer steps along the sampling path are sufficient.

The study further shows that accurate estimation of the BME is important for penalizing models with more complexity. By evaluating HM, TI, SS, and MOSS methods for a groundwater transport model selection problem, SS and MOSS show the most accurate results. Accurate estimation of the BME results in accurate penalization of more complex models. The estimated BME does not necessarily correlate with the true BME resulting in correlated underestimation or overestimation across all the candidate models, rather the estimation bias is a function of model complexity and model fidelity. In other words, estimation bias is method dependent. It is favorable to use an estimator which bias correlates with the true solution, since this will not change the model weights and ranking because we are interested in the BME ratios and not their absolute values (Equation (3)). In line with [23]), our empirical findings show that SS estimators have this property. It would be interesting to investiagate whether future studies would validate or refute these empirical findings.

**Author Contributions:** Conceptualization, A.S.E. and M.Y.; methodology, A.S.E.; software, A.S.E.; validation, A.S.E. and M.Y.; formal analysis A.S.E. and M.Y.; investigation, A.S.E. and M.Y.; resources, M.Y.; data curation, M.Y.; writing—original draft preparation, A.S.E.; writing—review and editing, A.S.E. and M.Y; visualization, A.S.E.; supervision, M.Y.; project administration, M.Y.; funding acquisition, M.Y.

**Funding:** The two authors were supported by the U.S. Department of Energy grant no. DE-SC0008272. The first author was also partly supported by the U.S. National Science Foundation award no. OIA-1557349. The second author was also partly supported by the U.S. Department of Energy grant no. DE-SC0019438 and U.S. National Science Foundation grant no. EAR-1552329.

**Acknowledgments:** We thank three anonymous reviewers for providing comments that helped to improve the paper. The data and computer codes used to produce this paper are available by contacting the corresponding author at mye@fsu.edu.

**Conflicts of Interest:** The authors declare no conflict of interest.

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
