# Peer review of "Making Steppingstones out of Stumbling Blocks: A Bayesian Model Evidence Estimator with Application to Groundwater Transport Model Selection"

_water, doi:10.3390/w11081579_

Round 1

Reviewer 1 Report

The paper investigates the use of BME to measure the accuracy of 4 selected groundwater transport models, by taking into account both the model capability in reproducing the physical process itself and the model complexity. Different numerical methods for BME calculation are tested and discussed. The experiments of Anamora et al., 1990 are used as benchmark for the scope. 

The topic is interesting, since often an overfitting of theoretical models occurs when few measurements are available for the training phase, especially when several input parameters are required. 

The paper is well written and structured, even if some oversights have to be corrected, e.g.: 

-L. 84: end point misses; 

-L. 84: 'such as' is repeated twice as well as 'we introduce' at L. 223; 

-L. 518-519: the sentence is incomplete; in some equation chinese letters appears, as in eqs. 23 and 25). 

I suggest the publication of the paper, after an overall proofreading of the text.

My main concern is about the use of a data set derived from only one experiment. The use of more measurements, (also under different boundary conditions) would increase the strengths of the results obtained in the present work. Did the authors test the BME calculation for accuracy estimation by comparing theoretical models with other observations? Moreover, it would be useful to investigate if the capability of BME to measure the model accuracy could be affected by the range variability of the input parameters.

Author Response

Please see the attached response letter. Thank you very much.

Reviewer 2 Report

Dear editor, 

I read the manuscript carefully. However, I should mention that the Bayesian statics and marginal likelihood are quite special topics which I have limited knowledge about. Assuming that the statistical and mathematical derivations are right and written flawlessly, which I expect from a two-author paper prepared by Dr. Ye, I found the entire methodology and conclusions fair and acceptable. However, I noticed numerous (minor) editorial and grammar issues, which at some points, made it difficult to read. The current version needs to be very carefully proofread since I am afraid similar typos and carelessness misrepresent the equations. All in all, from my viewpoint, this manuscript is acceptable for publication in your journal after minor revisions. 

Reviewer,

Author Response

Thank you very much for reviewing the article and for your positive evaluation. We agree that the article has several typos and grammatical errors. We did a thorough proof-reading and we corrected many langue errors as shown in the revised manuscript. Thank you very much.

Reviewer 3 Report

The work doesn't present an important hydraulic relavancy. I suggest to change journal and to improve the work that in the current state is unclear.

Author Response

Please see the attached report. Than you very much. 
